# On the effects of vendor balancing in deep learning for mammography

**Edwin D. de Jong**                                    EDWIN.DEJONG@SCREENPOINTMED.COM

**Jaap Kroes**                                          JAAP.KROES@SCREENPOINTMED.COM

*ScreenPoint Medical, Toernooiveld 300, 6525 EC Nijmegen,The Netherlands*

## Abstract

Machine learning initiatives in the medical domain are often restricted by the data that is available. In mammography, especially cancerous imaging data is typically difficult and costly to acquire. As a result, data imbalance plays a relatively major role, in contrast with general image recognition projects where large curated image databases are available. Quite some research exists on the *class imbalance* problem, which plays a role in many domains. Here, in contrast, we focus on an imbalance problem more specifically tied to the medical domain: *vendor imbalance*.

Various approaches for dealing with imbalanced data are available in general. Here, we report on a case study of the effect of over-sampling as an approach to deal with vendor imbalance. We consider CNN training for soft-tissue lesion detection in mammography. A sequence of over-sampling configurations are compared, representing a gradual shift from no balancing, where data from each vendor is sampled proportionally to its abundance, to full balancing, where all data is sampled uniformly. Contrary to our expectations, for this learning problem it is found that the average performance across the manufacturers is maximal when no balancing is used.

**Keywords:** Vendor imbalance

## 1. Machine learning in a multi-vendor context

In many machine learning applications, the source of the training data is not a factor of concern. When models are trained on the very large image databases that are available nowadays ((Deng et al., 2009), (Krasin et al., 2017), (Wu et al., 2019)), the differences between images taken with cameras from different vendors are relatively small, and the vast number of training samples from each camera type ensures that trained models can deal with images from most common camera types and brands.

For medical machine learning, this is different; there are distinct differences between the images produced by devices from different vendors. For example, mammography devices differ in terms of the X-ray spectra used through choice of X-ray tube target/filter combination, the digital detector technology, automatic exposure control (AEC), and image processing and presentation (Keavey et al., 2012). Given the cost and difficulty of obtaining medical image data for training, the number of available images per manufacturer, and per combination of manufacturer and class, can vary widely. These differences, combined with the vendor imbalance problem, complicate the training of CNNs for classification in several ways. First, the local optimum for different vendors may be reached at different points during the training. Furthermore, the global optimum for the different vendors is likely

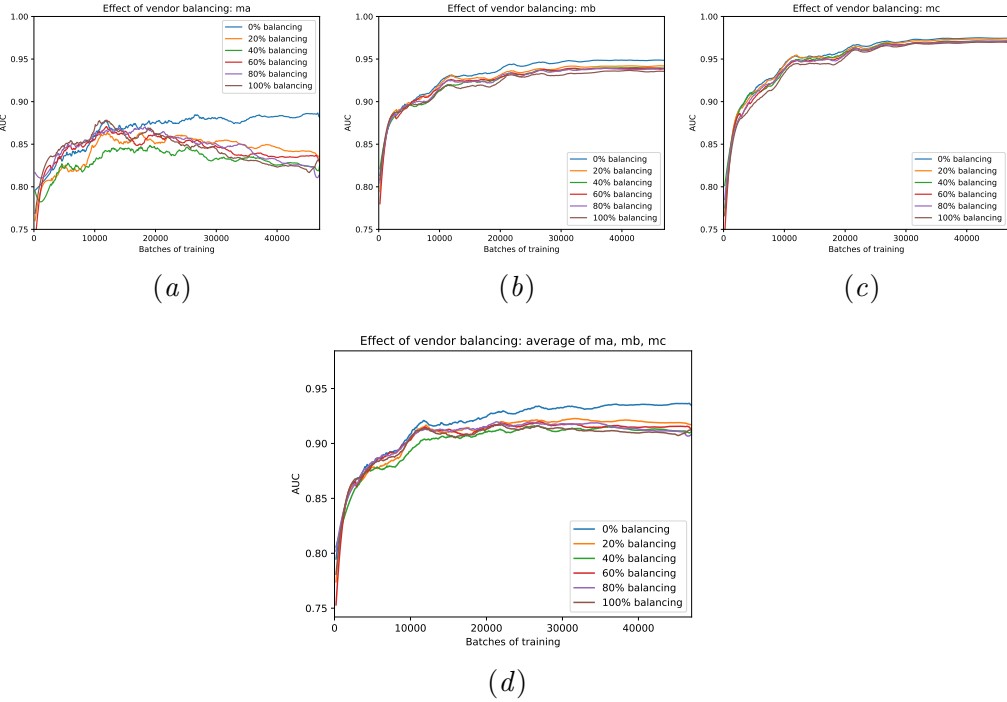

Figure 1: Validation performance (AUC) for vendors ma, mb, and mc (top) and averaged results (bottom). Each line represents a different oversampling configuration.

to differ, as data from different vendors has different characteristics and may thus require different features in the network.

The most conspicuous form of imbalance in medical machine learning is class imbalance, referring to an uneven distribution of available samples over the classes that are to be predicted. Class imbalance is a well studied problem within machine learning. A good overview of techniques used to address class imbalance in deep learning is provided in (Buda et al., 2017). This source describes and investigates several methods: oversampling, undersampling, two-phase training, and thresholding. Of these methods, oversampling was found to perform best. Other methods used for imbalance include GANs and augmentations.

## 2. Experimental setup

Here, we use oversampling to address vendor imbalance. We select a degree of oversampling for each vendor that specifies to what degree the available samples for a vendor will be oversampled during training. For example, choosing an oversampling factor of 1.5 for a particular vendor means that each sample from this vendor has a 50% higher probability of being selected for a batch of training.

The aim here is not to maximize performance, but to study the effect of over-sampling. We use a 13-layer VGG network with batch size 100. 80% of the available samples is used for training; the remaining 20% is used for validation. For augmentation, we use random

translation, random flipping, random rotation with 90 degrees, and cropping, all applied independently with probability 0.5.

We select samples from three different vendors, named manufacturer `ma`, mb and mc here. For details on the origin of the samples, the reader is referred to (Kooi et al., 2016). The amount of samples selected for each combination of vendor and class is as below. To analyze the effect of oversampling, we define several oversampling configurations, going from no oversampling to full balancing; see table 1.

| vendor | normal samples | abnormal samples | 0% | 20% | 40% | 60% | 80% | 100% |
|--------|----------------|------------------|------|------|-------|-------|-------|-------|
| ma | 1000 | 100 | 1.00 | 8.05 | 17.01 | 28.74 | 44.79 | 68.09 |
| mb | 20000 | 2000 | 1.00 | 1.16 | 1.37 | 1.64 | 2.01 | 2.55 |
| mc | 50000 | 5000 | 1.00 | 1.00 | 1.00 | 1.00 | 1.00 | 1.00 |

Table 1: Amount of samples selected per vendor and oversampling factors for configurations from no oversampling (0%) to full vendor balancing (100%). Factors differ slightly from the inverse of the numbers of samples per vendor due to case-based selection.

## 3. Results

To measure performance, we use the Area Under the Curve (AUC) of the Receiver Operating Characteristic (ROC) curve; see Figure 1. All curves are averages of five independent runs, and show the average AUC over time (number of training batches). For vendor `ma`, using 0% balancing is clearly the best option. For all other configurations, the performance initially (up to around batch 12000) improves, but then begins to deteriorate. The likely explanation is that the model then starts to over-train on vendor `ma`. For vendor `mb`, using no oversampling also gives the best performance. Here however, all configurations continue to make progress over time. For vendor `mc` finally, the difference between the various configurations is very small. The bottom figure shows the average of the auc calculated per vendor, so that small and large vendors contribute equally; i.e. the data is not pooled, as that would place a larger weight on vendors with more data. The behavior of this measure shows that not using balancing yields the best results, due to the detrimental effects of oversampling on the smaller vendors.

## 4. Discussion and Conclusion

This work reports empirical results on a particular machine learning problem in the context of lesion classification for mammography to gain insight into the issue of vendor imbalance. It was found that, contrary to expectation and to experiences in class imbalance problems, oversampling does *not* help in our situation to improve the performance for vendors with low amounts of data. It is important to realize that these findings cannot be assumed to generalize to all vendor-imbalance problems; it could well be that vendor imbalance in other (medical or non-medical) contexts can have other characteristics. Therefore, performing a similar analysis in other contexts would form a useful extension of this work.

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
