# OpenReview forum: "On the effects of vendor balancing in deep learning for mammography"
_MIDL.io/2019/Conference/Abstract — MIDL Abstract 2019_

### Official Review · AnonReviewer2 · 2019-04-30
**Interesting negative results, but experimental protocol too weak to ensure correctness of the results and needs more experiments.**

**Rating:** 2
**Confidence:** 2

**Review:**

The authors tackles the problem of vendor imbalance in mammography classification. More specifically, they test the
hypothesis that oversampling wrt the different vendors will improve overall performances.

The tested dataset has three vendors: ma (1100 samples, 1.4% of dataset), mb (22000 samples 28.2% of dataset) and mc (55000 samples, 70.4% of dataset). Surprisingly, they found that doing nothing is better that using oversampling, for any sampling factor.

Pros:
- Reporting negative results is important, and the authors stress that those results might not be generalizable to other tasks or areas.

Cons:
- My main concern is that oversampling should benefit vendor ma the most (AUC for ma increases, while AUC for mb and mc go down), yet the balancing decreases performances by a wide margin (from 0.88 to 0.80).
- I find surprising that the curves of the 6 different trainings configurations follow the same path (+- some value) for mb and mc. Since the sampling of the training images is supposed to be random, I would expect more variability on the performances over time..

Suggestions:
- Testing the balancing with different datasets would give much more information (since there is two levels of imbalance): ma+mc, ma+mb, mb+mc. My intuition is that trying balancing ma introduces more problems, due to the much higher imbalance (most of the batches comes from ma and very few from mb or mc).
- Still with ma+mb+mc, balancing only ma or only mb.
- Testing a balancing offline (almost the same, but not exactly in the details) could be another sanity check.

Minor:
- Mention explicitly that this is classification problem.
- The paper lacks a table with quantitative results.

---

### Official Review · AnonReviewer1 · 2019-05-01
**Good validation study of over-sampling in the context of data source imbalance**

**Rating:** 3
**Confidence:** 2

**Review:**

The abstract deals with a relatively untouched, but important concern in medical imaging - that of imbalance in training datasets in terms of the scanners used to acquire the images. In some cases, scanner characteristics can vary highly and this could lead to a model trained on one scanner to perform badly on images from another scanner.

In the dataset considered in this work, however, this does not seem to be the case. Over-sampling from the vendor with a small number of images leads to overfitting on that vendor's images. Thus, sampling proportionately to the number of available images from each scanner leads to the best results. However, as expected, higher performance is obtained on vendors with more images that those with fewer images.

Overall, the abstract presents a good validation study on an important problem in medical imaging analysis.

---

### Decision · Program_Chairs · 2019-05-06
**Acceptance Decision**

Accept